# Impact of 3MeV Energy Proton Particles on Mid-IR QCLs

**DOI:** 10.3390/nano13040677

**Published:** 2023-02-09

**Authors:** Petrişor Gabriel Bleotu, Laura Mihai, Dan Sporea, Adelina Sporea, Mihai Straticiuc, Ion Burducea

**Affiliations:** 1National Institute for Laser, Plasma and Radiation Physics, CETAL, 077125 Magurele, Romania; 2Horia Hulubei National Institute for R&D in Physics and Nuclear Engineering (IFIN-HH), 30 Reactorului St., 077125 Magurele, Romania; 3Doctoral School of Physics, University of Bucharest, 077125 Magurele, Romania; 4LULI-CNRS, CEA, Institut Polytechnique de Paris, Universite Sorbonne, Ecole Polytechnique, CEDEX, 91128 Palaiseau, France

**Keywords:** Quantum Cascade Lasers (QCLs), protons radiation, spectroscopy

## Abstract

This paper reports the results obtained for a distributed-feedback quantum cascade laser (DFB-QCL) exposed to different fluences of proton particles: 10^14^, 10^15^ and 10^16^ p/cm^2^. Dedicated laboratory setups were developed to assess the irradiation-induced changes in this device. Multiple parameters defining the QCL performances were investigated prior to and following each irradiation step: (i) voltage-driving current; (ii) emitted optical power-driving current; (iii) central emitting wavelength-driving current; (iv) emitted spectrum-driving current; (v) transversal mode structure-driving current, maintaining the system operating temperature at 20 °C. The QCL system presented, before irradiation, two emission peaks: a central emission peak and a side peak. After proton irradiation, the QCL presented a spectral shift, and the ratio between the two peaks also changed. Even though, after irradiation, the tunning spectral range was reduced, at the end of the tests, the system was still functional.

## 1. Introduction

The rapid development of QCLs contributed to their widespread use for spectroscopic applications in the mid-IR and THz spectral range, targeting atmosphere constituents’ studies, the measurement of planetary gases composition, astronomy, astrophysics, astrochemistry and spaceborne instrumentation [1,2,3,4,5,6,7,8,9,10,11,12]. The competing sources for this range are superlattice multipliers, which have a far lower output power but can cover the 0.1-to-1 THz range [13,14,15,16,17], which has a strong potential for medical diagnostics [18]. For this purpose, prior to the space mission, the degradation evaluation of materials, components and systems under various irradiation conditions is mandatory [19]. Damage could be expected on launched systems, related to electronic components and systems due to electrons and protons trapped in the two radiation belts surrounding the Earth [20], solar energetic particles (SEP) produced by solar flares and coronal mass ejections [21] and Galactic Cosmic Rays (GCR) composed of electrons (~2%), protons (~90%), high-Z elements (C, O, Fe, ~1%) and alpha particles (~9%) [22,23]. Different types of semiconductor lasers were tested under gamma-ray, electron beam, proton and neutron irradiation conditions. Early studies on the 200 MeV proton irradiation of multi-quantum well GaAs/GaAlAs laser diodes (emitting wavelength λ = 780 nm), at fluxes from 4 × 10^8^ to 1.5 × 10^10^ p/(cm^2^/s) and fluence steps of 1 × 10^12^, 5 × 10^l2^, l × l0^13^, 5 × 10^13^ and l × 10^14^ p/cm^2^, were carried out without any biasing (open-circuit or short-circuit) during the irradiation or by applying a bias of 18, 35 or 45 mA [24]. Additionally, it has been shown that high-temperature operation produces a high change in the slope efficiency in regard to the low temperature drive. The same temperature-related effect is present in the case of the threshold current variation with the proton fluence. The optical power degradation is more evident for high-temperature–low-current measurement conditions than for the low-temperature–high-current situations. These biasing conditions also affected the threshold current damage factor. In 2000, Barnes et al. reported the irradiation of VCSEL arrays by 53, 79 and 192 MeV (200 nA) proton beams, at a flux of 10^11^ p/(cm^2^/s), for the highest energy used [25]. A higher degradation of VCSEL’s parameters occurred when higher currents were applied, along with the temperature stress increase. For some samples, the threshold current doubled for a fluence of 2 × 10^14^ p/cm^2^, corresponding to a drop in the output power by a factor of eight. The effects of proton irradiation (50 MeV energy, 3 × 10^13^ n/cm^2^ fluence, at room temperature) on the operating parameters of different laser diodes (VCSEL, QW, DFB) based on AlGaAs–GaAs, InGaAsP–InP and InGaAs–GaAs emitting at λ = 650–1550 nm was studied by Johnston and his team [26]. Pre-irradiation and post irradiation tests referred to the measurement of electrical and optical characteristics, including the emission wavelength and the width of the output spectrum. Depending on the laser type, the degradation of the threshold current in relation to the increased fluence was linear or non-linear. The different behavior of laser diodes was also noticed regarding the fluence required to moderately modify (by 20–30%) the shift of the threshold current. Displacement damage associated with the emitted radiation wavelength shift was minimal, as the measured shift was of about 0.2 nm. 

VCSEL laser diodes were studied by Kalavagunta et al. under 2 MeV proton irradiation with a fluence from 2 5 × 10^12^ to 10^14^ p/cm^2^ [27]. They found a linear degradation of the threshold current with a damage factor of K1 = 2.77 × 10^15^ p/cm^2^. An increase in the leakage current occurred, and the change in the emission wavelength was Δλ/λ = 0.71%. Defect-limited mobility degradation as a result of the irradiation produced an increase in the device resistivity. 

Johnston and Miyahira studied the effects of 51 MeV protons on heterostructure laser diodes produced in different materials (AlGaInP, operating wavelength λ = 600–700 nm; AlGaAs, λ = 630–950 nm; InGaAs, λ = 900–1100 nm; InGaAsP, λ = 1100–1550 nm) by monitoring, before and after the irradiation, the optical power-forward current characteristics at three case temperatures (20, 30 and 40 °C). From these curves, they derived the changes in: (i) the threshold voltage, (ii) the temperature sensitivity of the threshold voltage and (iii) the efficiency slope [28]. For all the tested devices, the increase in the threshold voltage and the decrease in the efficiency slope were noticed. For the units operating at λ = 1550 nm, the temperature sensitivity of the threshold voltage diminished as the proton fluence increased to 6 × 10^13^ p/cm^2^. 

Ionizing and displacement damage effects of the gamma-ray (dose rate of 3.9 Gy/s, total dose of 9 MGy) and 36 MeV protons (fluences of 10^10^ up to 10^13^ p/cm^2^) on long-wavelength AlGaInAs/InP-based VCSEL lasers, emitting at λ = 1400 to 1700 nm, were reported by Van Uffelen and colleagues [29]. Pre-irradiation and post mortem tests were performed in relation to temperature effects (variation between 10 and 50 °C) on the devices’ optical power-driving current and voltage-driving current characteristics. The results indicated a decrease in the emitted optical power that was simultaneous with the degradation of the threshold current and slope efficiency. Proton irradiation mostly affected the threshold current, while photons exposure contributed to the decrease in the slope efficiency. 

Quantum well (QW) AlGaInP laser diodes emitting at λ = 665 nm were investigated in relation to the gamma irradiation dose and temperature [30,31]. Following the exposure to a gamma-ray to the total dose of 140 kGy, a degradation of the differential efficiency by 15.3% was noticed, while the threshold current increased from 23 mA to 31 mA. The voltage–current characteristics were modified slightly after the irradiation. The diodes degradation was mainly due to the increase in the operation temperature. 

A report on the proton (60 MeV energy, flux of 10^7^ p/(cm^2^/s), fluence of 2 × 10^10^ p/cm^2^) and gamma-ray (dose rate 5.5 Gy/h, total dose 1 kGy) effects on mid-IR-emitting (λ = 2100 nm) GaSb-based DFB lasers intended to be used in space applications indicated no degradation of the specific curves examined: (i) optical power-driving current, (ii) current–voltage, (iii) wavelength as a function of current and temperature [32].

One commercially available VCSEL InGaAsP/InP (DFB) laser (λ = 1310 nm) and one edge emitter AlGaAs/GaAs Distributed Bragg Reflecting (DBR) laser (λ = 850 nm) were irradiated by 3 MeV protons at a fluence of 3 × 10^12^ p/cm^2^. As the proton beam was directed perpendicularly on the laser output facet, the VCSEL device was irradiated parallel to the junction plane, while the other device had the junction irradiated perpendicularly [33]. Off-line measurements of the lasers’ driving currents, forward voltage and monitoring photodiode current were performed. Following the proton irradiation, the lasers’ threshold current increased and its efficiency slop decreased. The AlGaAs/GaAs Distributed Bragg Reflecting laser was more sensitive to proton irradiation, having a pronounced degradation of the output power.

There are very few reports concerning the study of QCLs degradation upon irradiation. Recently, Fabry–Perot (FP) QCLs, having a central emission wavelength between 5300 and 8200 nm, were investigated either under high-energy proton irradiation (64 MeV) or after gamma-ray (60Co) irradiation, up to the maximum total doses between 200 and 463 Gy (Si). Before and following the irradiation, the laser was measured under quasi-CW operating conditions, as it concerns: (i) the emitted optical power (steps: 6, 12, 18, 24, 29 mW) at specific driving current levels; (ii) the threshold current; (iii) the slope of the emission efficiency. A FTIR spectrometer was used to monitor the emission spectral characteristics. The changes in the threshold current and the efficiency slope variation demonstrated that such lasers are appropriate for use under space radiation conditions [34]. Additional data on FP QCLs’ tests for space missions’ qualification can be found in a report by Bernacki et al. [35], where the degradation of the C-mount and the lasers’ front facet after irradiation and multiple handling operations is mentioned. 

This paper reports the evaluation of proton irradiation effects on a DFB-QCL in order to assess their possible use in mid-IR spectroscopy for gas tracing, as a part of future space missions operating in high-radiation environments. Even though, very recently, less expensive detection systems were developed based on the NIR diode laser [36,37,38], which exhibited sensitivities up to ppb, for this study, a DFB- QCL was selected due to its similarity with the tunable laser system already selected for the planned JUICE mission (Jupiter exploration, which is planned to launch this year). Additionally, detection systems based on QCLs are still one of the most sensitive setups, recently demonstrating detection levels up to ppt [39]. The novelties of this study refer to: the investigation of a DFB QCL subjected to proton irradiation considering multi-parameters evaluation characterizing the tested system: electrical, optical and electro-optic characteristics were measured as a function of the proton fluence, driving current and case temperature.

## 2. Materials and Methods

This study targeted the evaluation of DBF-QCL degradation under proton irradiation in order to estimate the possible use of such devices in mid-IR spectroscopy for gas tracing, as part of spaceborne equipment operating under severe radiation environments. The focus was methane detection during space missions, so the tested QCL for this work has a central emission wavelength of λ = 7550 nm. The device characteristics before irradiation were: the maximum optical power, −80 mW@25 °C; the wavelength tuning range, −7542 to 7553 nm; the threshold current, −110 mA, having a C-mount-type case. It was delivered without a window. For this reason, careful handling was required to avoid dust or skin oil contamination or the mechanical degradation of the laser facet. Latex gloves were used to manipulate the device when handling between the testing and irradiation processes, and its mounting on the active laser mount (ALM) was performed in a clean room. The ALM has a very fast ZnSe lens (working distance 0.7 mm) with XYZ degrees of movement for optimizing the beam delivery during the characterization. Precautions were also considered in mounting the C-mount on the ALM and fixing the laser’s terminals to the heat sink. 

The QCL was subjected to several subsequent proton irradiations at the 3 MV TandetronTM accelerator operated by “Horia Hulubei” National Institute of Physics and Nuclear Engineering personnel. Samples, mounted on a three-axis goniometer with a precision of 0.01°, were exposed to the 3 MeV proton beam at a normal incidence to the exit facet. The entire setup was operated inside a vacuum chamber at a pressure of 2 × 10^−7^ mbar. The setup can be found in [40]. The proton beam diameter for the uniform dose was 3 mm. The QCL was irradiated in three steps at fluences of 10^14^, 10^15^ and 10^16^ p/cm^2^ by keeping the beam current at 5 nA. Off-line measurements were conducted in the laboratory for the QCL before and after each irradiation, and the following characteristics of the QCL were monitored each time: (i) driving current-voltage; (ii) emitted optical power-driving current; (iii) central emitting wavelength-driving current; (iv) emitted spectrum-driving current; (v) transversal mode structure-driving current. All data were acquired for three QCL case temperatures: 10, 20 and 30 °C. The QCL’s characterization setup is depicted in Figure 1. A complete description of the automatized characterization of several types of QCLs, using a LabView interface, can be found in Bleotu et al. [41].

The QCL mounted on an active laser mount (ALM) was operated through an LD/TEC controller, model ITC4005QCL, from Thorlabs. All tests ran the QCL in the CW mode, for a driving current modification from 0 to 250 mA, determining the voltage–current curves. The QCL beam passes through an LWIR-AR-coated collimating lens (NA = 0.85) mounted on an XYZ translation stage. The beam is free-space coupled at the inputs of the measuring equipment by a Ge lens with a focal length of 70 mm. The optical power was monitored by an Ophir Nova II display connected to the 3A-FS detector head (spectral range—190–20,000 nm; clear aperture—9.5 mm; power noise level—4 μW; power linearity—± 1.5 %; power accuracy—± 8%; 30 min; maximum thermal drift—30 μW). The Bristol Instruments 721B-XIR (spectral range from 2000 to 12,000 nm; absolute accuracy—± 1 ppm, ±0.0008 nm @ 1000 nm; standard spectral resolution—12 GHz; S/N > 30 dB) was used to monitor both the emitted spectrum and the central wavelength. QCL beam analysis was performed with the Ophir/Spiricon Pyrocam III (sensitivity wave-length ranges from 1.06 to 3000 μm; 124 × 124 elements; pixel size 85 μm × 85 μm; pixel spacing 100 μm × 100 μm; LiTaO_3_ sensing material). The LD/TEC controller, the power meter and the optical spectrum analyzer/wavelength meter were controlled using the USB connection, while the Pyrocam III was employed in connection with a firewire interface.

## 3. Results

QCL performances were tested prior to and after the exposure to three different protons fluences: 10^14^ p/cm^2^; 10^15^ p/cm^2^ and 10^16^ p/cm^2^, and their results are presented in the following paragraphs. First, the QCL was ramped over 15 current levels above the threshold, and the current–voltage curve was registered for each step (Figure 2a–c), with no visible changes after irradiation. Figure 2d shows some changes in the QCL performances with the increasing radiation dose, especially for temperatures above room temperature (22–25 °C). To better understand the results of the threshold current variation with proton doses, we used a linear fit, and the results suggested that, after irradiation with the highest fluence, the threshold current decreases with 19.32 mA (16.85%), 21.68 mA (19.08%) and 25.37 mA (19.58%) for temperatures of 10 °C, 20 °C, and 30 °C, respectively. The corresponding measured slopes were −1.93212 × 10^−15^ mA/(p/cm^2^), −2.16778 × 10^−15^ mA/(p/cm^2^) and −2.53743 × 10^−15^ mA/(p/cm^2^). The decrease in the threshold current and the increase in the slope efficiency could suggest an improvement in the device performances. To determine the slope efficiency, we also analyzed the output power variation with the driving current before and after irradiation with the highest fluence, and the results are represented in Figure 3. The output optical power values measured before irradiation using a very sensitive thermopile were between 2.7 and 44 mW for driving current values variation between 110 and 220 mA. After 220 mA, the optical power saturates. After irradiation with a proton fluence of 10^16^ p/cm^2^, the optical power values increased around 10 mW for corresponding driving current values between 100 and 170 mA. After 170 mA, the device optical power results suggest the device degradation (Figure 3b). From the linear fit applied to the optical power versus driving current graph, an increase of 0.05 W/A was obtained for the slope efficiency at 10 °C, and decreases of 0.05 W/A and 0.07 W/A for 20 °C and 30 °C, respectively, were applied to the device case. 

The equivalent serial resistance was calculated for each step-in order to determine the changes in the device electrical performances (Figure 4a). Representing this parameter in relation to the case temperature and fluence variation (Figure 4b), an increase of about 3.4 Ω (17.65%), 3 Ω (16%) and 2.3 Ω (12.34%) was observed for temperatures of 10 °C, 20 °C and 30 °C, respectively, between the system before the exposure to radiation and its exposure to the highest fluence applied. This increase in the serial resistance implies a higher loss of the injected electrical power as heated, as also noticed in Figure 2d, and this issue can be removed by the cooling system. 

The QCL spectral tuning performances have also been tested prior to and after the radiation exposure, and the results obtained for the temperature of 10 °C are represented in Figure 5. Increasing the driving current between 100 and 250 nm, with a step of 10 mA, resulted in a primary emission mode and a secondary mode. Before irradiation, the QCL wavelength was tuned in the spectral range of 7535.6–7552.4 nm (16 nm, 15 emission lines, FWHM of around 0.0005 nm), increasing the driving current in the range of 110–250 mA, with a step of 10 mA (Figure 5a). After irradiation with a proton fluence of 10^16^ p/cm^2^, the spectral range was reduced to 7534.6–7549.3 nm (14.7 nm, 12 emission lines, FWHM of around 0.0009 nm, driving currents between 100 and 220 mA) (Figure 5b). Additionally, the signal-to-noise ratio decreased after irradiation, and the secondary mode gained more power (Figure 5b). As a next step, the wavelength shift was investigated. Prior to any irradiation session, the QCL emission was single-line, as the spectrum had a primary emission line (λ1) and a much smaller side line (λ2) (Figure 5). A laser stability, without emission mode hopping during the current or temperature tuning, guarantees a reliable use of QCL in mid-IR spectroscopy. After proton irradiation instead, the ratio of the amplitudes corresponding to the central (λ1) and the side peaks (λ2) changed, as can be noticed in Figure 6a–c, for all three temperatures.

For the lowest fluence (10^14^ p/cm^2^), the central emission wavelength shift (λ1) increases monotonically with the driving current for all temperatures (Figure 6), as compared with the other two fluences, where a revers phenomenon is noticed. This behavior is reflected by the degradation of the side-mode suppression ratio (SMSR) from an almost flat response for the device before irradiation towards a bent surface for the irradiated QCL (Figure 7). For a fluence of 10^16^ p/cm^2^, the minimum value of the SMSR is achieved at the temperature of 10 °C (Figure 7a), while for higher case temperatures, the two emission peaks are almost equal (Figure 7b,c). 

The QCL’s beam quality and lower fluences (e.g., 10^14^ to 10^15^ p/cm^2^) provide quite similar changes of the X and Y widths (Figure 8). An increase in the fluence to 10^16^ p/cm^2^ also produces an increase in the beam diameters across X and Y (Figure 8).

## 4. Conclusions

In this paper, we present the results obtained for the degradation of a DFB Quantum Cascade Laser when exposed to 3MeV proton radiation for three different radiation fluences (10^14^ p/cm^2^, 10^15^ p/cm^2^, 10^16^ p/cm^2^). The investigations were focused on the changes produced by proton exposure in a DFB QCL as it concerns the electrical and optical performances, such as: electrical characteristics (voltage-current—Figure 2, equivalent serial resistance—Figure 4), the degradation of the QCL’s single-mode emission spectrum (Figure 5), the change induced by irradiation in the central wavelength shift with the current/temperature (Figure 6) and the modification of the QCL’s SMSR following proton irradiation (Figure 7). The influence of proton irradiation on the emitted optical power (Figure 3) and the variation in the QCL beam quality as a function of the proton fluence (Figure 8) have been also tested.

The results showed that proton fluences of 10^16^ p/cm^2^ produced changes in the QCL optical signal, a spectral shift and optical power. 

More specifically, the threshold current decreased after exposure to 10^16^ p/cm^2^ with 19.32 mA (16.85%), 21.68 mA (19.08%) and 25.37 mA (19.58%) for temperatures of 10 °C, 20 °C and 30 °C, respectively, which correspond to measured slopes of −1.93212 × 10^−15^ mA/(p/cm^2^), −2.16778 × 10^−15^ mA/(p/cm^2^) and −2.53743 × 10^−15^ mA/(p/cm^2^). From the optical power dependence on the driving current variation, we obtained a very small decrease of around 0.06 mW/mA in slope efficiency for temperatures above room temperature and an increase of 0.05 mW/mA for the cooled setup (10 °C). These potential variations could be related to changes in the thermal contact between the QCL and the C- mount due to the dismounting and remounting procedure applied to QCL before each irradiation session [42]. 

Serial resistance increased by about 3 Ω after irradiation, suggesting a greater loss of the injected electrical power when heated up to 30 °C. Before the QCL exposure to proton radiation, the serial resistance was more stable to temperature variations, having a standard deviation of 0.19 Ω compared with the irradiated case, when the standard deviation was up to 0.66 Ω. 

After the exposure to a proton fluence of 10^16^ p/cm^2^, the beam quality presented a degradation after irradiation (similar effect observed in [43]), which is represented by the increase in the FWHM values of the emission lines, from 0.5 nm to around 1 nm (Figure 5), but also from investigation of beam profile (Figure 8). These changes can be related to the laser emission modes hopping, which also modified the single-mode suppression ratio (Figure 7) when a driving current above 220 mA was applied to the laser diode. 

During the in-vacuum proton beam irradiation, even at 5 nA, thermal transfer will occur, which may increase the temperature locally by tens or even hundreds of degrees for a high dose. The protons can induce internal defects within the QCL structure (hundreds of thin (nm scale) layers) due to atomic displacement, thus degrading the electronic band structure [34]. Thermal stress represents a well-known source of defect generation in crystals and semiconductors, which may explain the changes seen in the emission band, similar to the results reported in Ref. [44]. Considering all of this, we may conclude that, even though the QCL was affected in a small percentage by the exposure to proton radiation fluences up to 10^16^ p/cm^2^ and an energy of 3 MeV, it was still functional at the end of experiment, suggesting its possible use for space or hazardous environment applications, where similar irradiation fluences are used. 

All these results can contribute to the development of an extensive database covering different aspects of proton irradiation effects on mid-IR QCL, which could be of interest for researchers involved in: (i) the investigation of the ionizing radiation impact on mid-IR semiconductor emitters, (ii) applications of THz technology, (iii) the design of instrumentation operating in harsh environments (i.e., spaceborne equipment).

## Figures and Tables

**Figure 1 nanomaterials-13-00677-f001:**
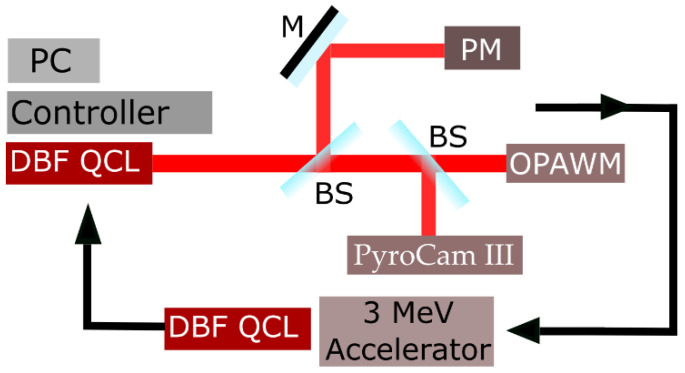
The sketch of the QCL testing setup, including: the current and temperature controller; DBF-QCL; PyroCam III; Power meter (PM); Optical spectrum analyzer and wavelength meter (OSAWM); Computer—PC; Mirror (M) and two Beam-splitters (BS). After the initial characterization, the QCL was irradiated using the 3 MeV accelerator, followed by another set of characterizations.

**Figure 2 nanomaterials-13-00677-f002:**
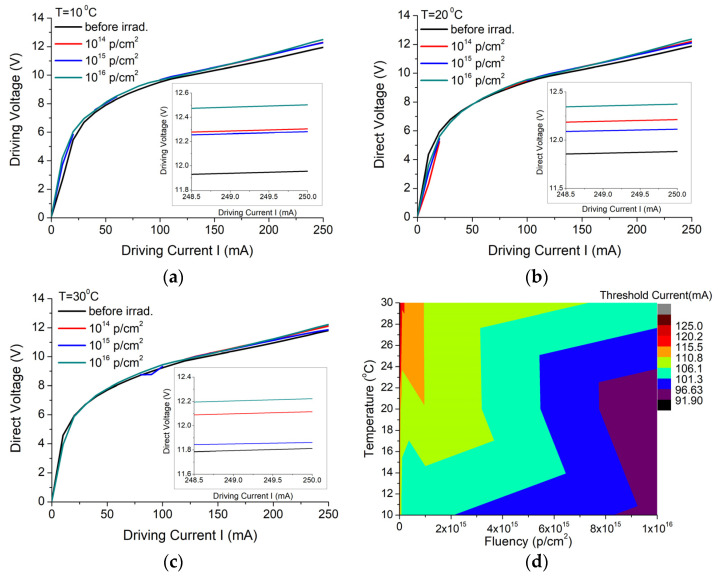
Voltage–current characteristic dependence on the proton fluence, measured at (**a**) 10, (**b**) 20 and (**c**) 30 °C case temperatures, and (**d**) 3D representation for the current threshold dependence on the temperature variation and p radiation fluences.

**Figure 3 nanomaterials-13-00677-f003:**
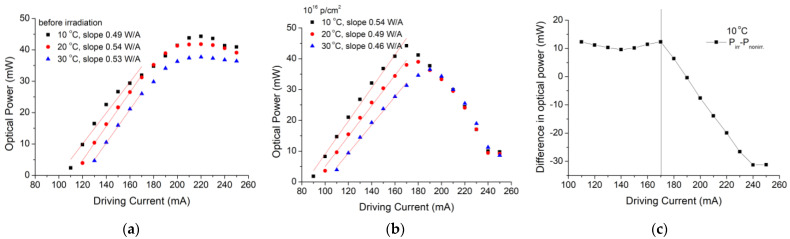
The emitted optical power vs. the driving current for three case temperatures (**a**) before irradiation and (**b**) after 10^16^ p/cm^2^ irradiation; (**c**) difference between the optical power before and after irradiation corresponding to the device cooled at 10 °C.

**Figure 4 nanomaterials-13-00677-f004:**
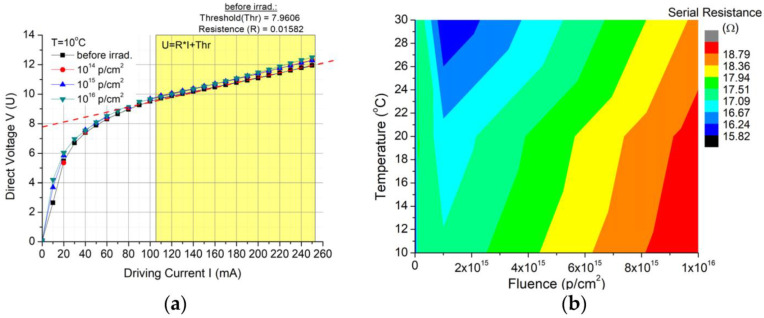
(**a**) Definition of the device serial resistance and (**b**) serial resistance variation with the radiation fluence and QCL case temperatures.

**Figure 5 nanomaterials-13-00677-f005:**
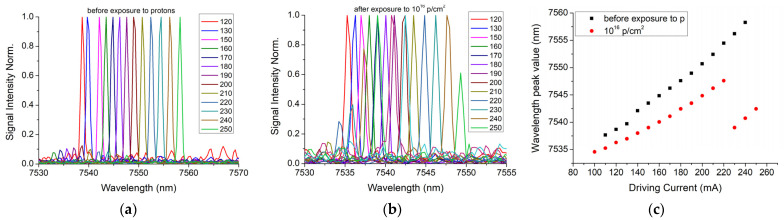
QCL tuning capabilities with the driving current variation for a case temperature of 10 °C (**a**) before irradiation, (**b**) after exposure to 10^16^ p/cm^2^; (**c**) the peak wavelength dependence on the driving current.

**Figure 6 nanomaterials-13-00677-f006:**
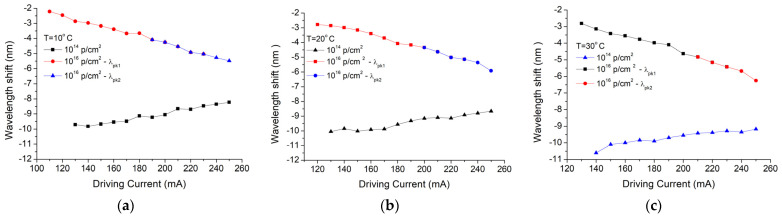
Protons irradiation impact (fluences of 10^14^ p/cm^2^ and 10^16^ p/cm^2^) on the driving current-induced central wavelength (λ1) shift for: (**a**) 10 °C, (**b**) 20 °C, (**c**) 30 °C.

**Figure 7 nanomaterials-13-00677-f007:**
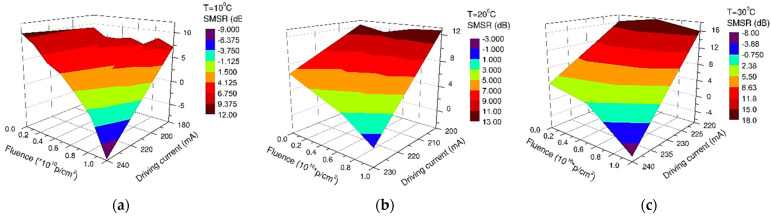
Decrease in the SMSR with the irradiation and driving current for the operating temperatures: (**a**) 10 °C, (**b**) 20 °C, (**c**) 30 °C.

**Figure 8 nanomaterials-13-00677-f008:**
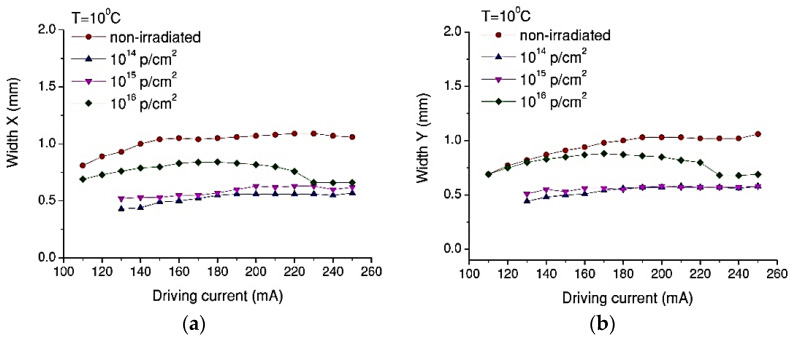
Beam width along the X (**a**) and Y (**b**) axis function of the driving current for Tc = 10 °C and different fluences.

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
