# Peer review of "Impact of 3MeV Energy Proton Particles on Mid-IR QCLs"

_nanomaterials, 2023, doi:10.3390/nano13040677_

Round 1

Reviewer 1 Report

Manuscript reference number: nanomaterials-2156575

Title: Impact of 3MeV energy proton particles on mid-IR QCL’s

Authors: Gabriel Petrisor Bleotu et al

In this manuscript the authors discussed the results obtained for a distributed-feedback quantum cascade laser exposed to different fluencies of proton particles. The paper is rather interesting and well organized and could be published on this journal but after several revisions according to the list of the following critical points:

a)      Conclusions should be modified. The Conclusions should only mention the obtained results and not the name of the project, which could be regularly mention in the acknowledgment of funding sections.

b)     There are several mechanical deficiencies of the figures: Figures 2,3 and 5 cannot be read due to the small size of the axis, labels etc…We strongly suggest to increase the size to improve the readability, or to re-organize or embody some figures (i.e. in figs.2). Moreover in Fig.5 there are too many curves.

Author Response

We would like to thank reviewer for his valuable comments that are very helpful for revising and improving our paper, as well as the important guiding significance to our researches. We have studied comments carefully and have made correction which we hope meet with approval. All responses are presented below.

Point 1: Conclusions should be modified. The Conclusions should only mention the obtained results and not the name of the project, which could be regularly mention in the acknowledgment of funding sections.

Response 1: We rephrased as suggested, excluding the project name. The initial phrase was replaced with:

“In this paper we present the results obtained for the degradation of a DFB Quantum Cascade Lasers when exposed to 3MeV proton radiation and for 3 different radiation fluencies (1014 p/cm2, 1015 p/cm2, 1016 p/cm2).”

Point 2: There are several mechanical deficiencies of the figures: Figures 2,3 and 5 cannot be read due to the small size of the axis, labels etc…We strongly suggest to increase the size to improve the readability, or to re-organize or embody some figures (i.e. in figs.2). Moreover in Fig.5 there are too many curves.

Response 2: All figures have been modified as suggested. We did not remove from the Figure 5 curves due their importance and due to the other reviewers’ comments. We changed the color of the curves and enlarged all figures, to be clearer. Figures that were changed can be found in the attached file.

Reviewer 2 Report

Quantum Cascade Lasers are a class of lasers well known from many years but that recently have been considered as a promising tool for compact mid-IR lasers with multiple possibilities of application including space. Therefore the analysis of the resistance of such devices to ionizing radiation seems to me very interesting. This research is not new, many teams and projects are in progress towards that goal. So the paper alings to this broad research line and can attract interest. 

The irradiation source choice is reasonable and the dosimetry is clearly explained for reproduction in other irradiation facilities with few MeV protons. 

Results are clear, methods are explained and there is a clear possibility of reproducing the results in other labs. So the paper can be accepted for publication in its present form. 

I would optionally to comment a bit more on a few sentences that could help to increase the readability of the manuscript to a broader audience. For example authors wrote in the conclusion: "Serial resistance increased with about 3 Ω after irradiation, suggesting a higher loss 305 of the injected electrical power when heated."This sentence in my opinion needs to be worked out a bit more. What has been the heating? The same heating but without proton irradiation would have result in the same resistance increasing? All this is explained in the main text (fig 4) but a bit confusing in the short sentence in the conclusion

Authors indicate that the spectral range of laser emission after irradiation has been reduced in one nm out of the 16 nm bandwidth after proton irradiation. Do they have an explanation for that? Emission bandwidth is a very important parameter to analyse the possibility of short pulse emission, so more indications on how this bandwidth can be reduced as a function of the irradiation dose. 

All that are optional minor points.

Author Response

We thank the reviewer for his careful reading of the manuscript and constructive remarks. We have taken the comments on board to improve and clarify the manuscript. Please find below a detailed point-by-point response to all comments.

Point 1: I would optionally to comment a bit more on a few sentences that could help to increase the readability of the manuscript to a broader audience. For example authors wrote in the conclusion: "Serial resistance increased with about 3 Ω after irradiation, suggesting a higher loss of the injected electrical power when heated."This sentence in my opinion needs to be worked out a bit more. What has been the heating? The same heating but without proton irradiation would have result in the same resistance increasing? All this is explained in the main text (fig 4) but a bit confusing in the short sentence in the conclusion

Response 1:

The sentence "Serial resistance increased with about 3 Ω after irradiation, suggesting a higher loss of the injected electrical power when heated." has been replaced by:

“Serial resistance increased with about 3 Ω after irradiation, suggesting a greater loss of the injected electrical power when heated up to 30 °C. Before QCL exposure to proton radiation, the serial resistance was more stable to temperature variations, having a standard deviation of 0.19 Ω compared with irradiated case, when the standard deviation was up to 0.66 Ω. “

Point 2: Authors indicate that the spectral range of laser emission after irradiation has been reduced in one nm out of the 16 nm bandwidth after proton irradiation. Do they have an explanation for that?

Emission bandwidth is a very important parameter to analyse the possibility of short pulse emission, so more indications on how this bandwidth can be reduced as a function of the irradiation dose.

Response 2: We would like to thank to the reviewer for this important observation. We investigated the reduction of the bandwidth and we add to our previous explanation the following:

In the manuscript we specified at lines 310-322:

“After exposure to proton fluence of 1016 p/cm2, the beam quality presented a degradation after irradiation (similar effect observed in [43]) that is represented by the increase in the FWHM values of the emission lines, from 0.5 nm to around 1 nm (Figure 5), but also from investigation of beam profile (Figure 8). These changes can be related to the laser emission modes hopping, that modified also the single-mode suppression ratio (Figure 7), when driving current above 220 mA was applied to the laser diode.

During the in-vacuum proton beam irradiation, even at 5 nA, thermal transfer will occur, that may increase the temperature locally by tens or even hundreds of degrees - for high dose. The protons can induce internal defects within the QCL structure (hundreds of thin (nm scale) layers) due to atomic displacement thus degrading the electronic band structure [44]. Thermal stress represents a well-known source of defect generation in crystals and semiconductors, that may explain the changes seen in the emission band, similar to the results reported in Ref. [45].”

The following reference has been added:

  1. Sin, Y., Lingley, Z., Brodie, M., Presser, N., Moss, S., Kirch, J., Chang, C.-C., Boyle, C., Mawst, L.J., Botez, D., Lindberg, D., Earles, T., 2015. Destructive physical analysis of degraded quantum cascade lasers. Proceedings of SPIE - The International Society for Optical Engineering 9382. https://doi.org/10.1117/12.2076641
  2. Tanya L. Myers, Bret D. Cannon, Carolyn S. Brauer, Stewart M. Hansen, and Blake G. Crowther, "Proton and gamma irradiation of Fabry–Perot quantum cascade lasers for space qualification," Appl. Opt. 54, 527-534 (2015)
  3. Evans, C.A., Jovanovic, V.D., Indjin, D., Ikonic, Z., Harrison, P., 2006. Investigation of thermal effects in quantum-cascade lasers. IEEE Journal of Quantum Electronics 42, 859–867.

Reviewer 3 Report

In this paper, a distributed-feedback quantum cascade laser (DFB-QCL) was exposed to different fluencies of proton particles, and a dedicated laboratory setups were developed to assess the irradiation induced changes of QCL performance, such as electrical characteristics, emission spectrum, central wavelength shift, the emitting optical power and laser beam quality, etc. The paper is generally well written and organized; the experimental results are completely presented. The paper is a interesting work and suitable for publication on Nanomaterials. However, I would like the authors to consider a few points when polishing up the final draft.

1) Since the first demonstration in 1994 (Science 264:553-556,1994), progress in the development of quantum cascade lasers (QCLs) has been breathtakingly rapid. Some in-depth review of recent advances in QCL technology and applications have been reported, such as Applied Spectroscopy Reviews, 48:523–559, 2013; Optics Express 23:5167-5182,2015;Chemical Society Reviews 46:5903,2017, the related publications mentioned here or references therein can be used for discussion in the revised paper. 

2) Since methane (CH4) gas was selected for the JUICE space mission, so the mid-infrared (MIR) QCL with a central emission wavelength at λ = 7.55 μm was used and tested in this work. Comparising to MIR QCLs, near-infrared (NIR) diode lasers have been developed quite mature and has higher spectral characteristics. Ppm-ppb level detection sensitivity of CH4 can be easily achieved by combining various sensitive detection techniques. Why an expensive QCL laser was selected without considering the cost-effective NIR diode laser, for example, Sensors and Actuators B 220:1000-1005,2015; ACS Sensors 5: 36073616,2020; Optics and Lasers in Engineering 151: 106907,2022; Sensors and Actuators B369:132234,2022. Please make a simple comparison and discussion properly in the revised paper.

3) Line 235: Increasing the driving current between 100 and 250 nm, with a step of 10mA,.... The units (nm Vs mA) of the driving current and step is inconsistent, please check this issue.

4) Frequency noise characteristics is also a key factor affecting laser performance, please discuss this issue appropriately with the laser used in the experiment as an example.

5) For uniform comparison, I suggest that the units of emission spectral range is changed from microns (μm) to nanometers (nm).

6) The QCL mounted on an active laser mount (ALM) was operated through a LD/TEC controller from Thorlabs with the following performances: .....

If the LD/TEC controller model is provided, then the tedious technical parameters of the instrument are completely omitted for the readers.

Author Response

We thank the reviewer for his very helpful comments and suggestions, which we have fully addressed point by point in the paper as following:

Point 1: Since the first demonstration in 1994 (Science 264:553-556,1994), progress in the development of quantum cascade lasers (QCLs) has been breathtakingly rapid. Some in-depth review of recent advances in QCL technology and applications have been reported, such as Applied Spectroscopy Reviews, 48:523–559, 2013; Optics Express 23:5167-5182,2015; Chemical Society Reviews 46:5903,2017, the related publications mentioned here or references therein can be used for discussion in the revised paper.

Response 1: The suggested publications are indeed significant in the wider topic area of the paper, thus we included all these as references:

  1. J. Faist, F. Capasso, D. L. Sivco, A. L. Hutchinson, C. Sirtori, and A. Y. Cho, ‘‘Quantum cascade laser,’’ Science 264,553–556 (1994).
  2. Li, J. S., Chen, W., and Fischer, H. (2013) Quantum cascade laser spectrometry techniques: A new trend in atmospheric chemistry. Appl. Spectrosc. Rev. 48(7): 523–559. doi:10.1080/05704928.2012.757232.
  3. Miriam Serena Vitiello, Giacomo Scalari, Benjamin Williams, and Paolo De Natale, "Quantum cascade lasers: 20 years of challenges," Opt. Express 23, 5167-5182 (2015).
  4. A. Schwaighofer, M. Brandstetter and B. Lendl, Chemical Society Reviews, 2017, 46, 5903-5924.

Point 2: Since methane (CH4) gas was selected for the JUICE space mission, so the mid-infrared (MIR) QCL with a central emission wavelength at λ = 7.55 μm was used and tested in this work. Comparising to MIR QCLs, near-infrared (NIR) diode lasers have been developed quite mature and has higher spectral characteristics. Ppm-ppb level detection sensitivity of CH4 can be easily achieved by combining various sensitive detection techniques. Why an expensive QCL laser was selected without considering the cost-effective NIR diode laser, for example, Sensors and Actuators B 220:1000-1005,2015; ACS Sensors 5: 3607−3616,2020; Optics and Lasers in Engineering 151: 106907,2022; Sensors and Actuators B 369:132234,2022. Please make a simple comparison and discussion properly in the revised paper.

Response 2: The primary reason for selecting a DFB-QCL system for this study was the power and emission capabilities of these laser types at ambient temperature, which were similar to those of the tunable laser system (TLS), type DFB-ICL, selected for JUICE space mission. In addition, ppb-level detection capabilities similar to those of a QCL have been demonstrated very recently for NIR laser diode systems (e.g., paper https://doi.org/10.1016/j.pacs.2022.100353; Optics and Lasers in Engineering 151: 106907,2022; Sensors and Actuators B 369:132234,2022).

Still, detection systems based on QCLs demonstrated to be one of the most sensitive, recently demonstrating detection levels up to ppt (https://doi.org/10.1364/OE.434128 ).

The following text has been introduced in text:

“Even though very recently there were developed less expensive detection systems based on NIR diode laser [36-38], that proved sensitivities up to ppb, for this study was selected a DFB- QCL due to its similarity with the tunable laser system already selected for the planned JUICE mission (Jupiter exploration, planned to launch this year). Also, detection systems based on QCLs there are still one of the most sensitive setups, recently demonstrating detection levels up to ppt [39].”

Point 3: Line 235: Increasing the driving current between 100 and 250 nm, with a step of 10mA,.... The units (nm Vs mA) of the driving current and step is inconsistent, please check this issue.

Response 3: We have corrected all units to be consistent.

Point 4: Frequency noise characteristics is also a key factor affecting laser performance, please discuss this issue appropriately with the laser used in the experiment as an example.

Response 4: We would like to thank to the reviewer for this important observation. During our experiment the frequency noise characteristics have not been quantified. The only possible quantification can be retrieved, at this time, from figure 5 a and b. We have understood the importance of frequency noise in our research and we intend to proper qualify this noise, taking into account your well received advice, in our future work.

The main reasons of frequency noise can be related to the internal noise of the QCL, temperature and current driver [ https://ieeexplore.ieee.org/stamp/stamp.jsp?tp=&arnumber=5896007] and the effects of the protons irradiation. The increase in the temperature, due to the high proton fluences, has been induced atomic displacement and thermal stress [45] which may cause the frequency noise.

The following text has been added in the conclusion section:

“During the in-vacuum proton beam irradiation, even at 5 nA, thermal transfer will occur, that may increase the temperature locally by tens or even hundred of degrees - for high dose. The protons can induce internal defects within the QCL structure (hundreds of thin (nm scale) layers) due to atomic displacement thus degrading the electronic band structure [44]. Thermal stress represents an well-known source of defect generation in crystals and semiconductors, that may explain the changes seen in the emission band, similar to the results reported in Ref. [45]. These effects have been observed in the increase of the frequency noise from Figure 5a and b.”

Point 5: For uniform comparison, I suggest that the units of emission spectral range is changed from microns (μm) to nanometers (nm).

Response 5: We have corrected all units to nm scale for the emission spectral range.

Point 6: The QCL mounted on an active laser mount (ALM) was operated through a LD/TEC controller from Thorlabs with the following performances: .....

If the LD/TEC controller model is provided, then the tedious technical parameters of the instrument are completely omitted for the readers.

Response 6: The technical parameters were erased and the controller model included in the text as following:

“The QCL mounted on an active laser mount (ALM) was operated through a LD/TEC controller, model ITC4005QCL, from Thorlabs with the following performances: current control range – up to 5 A; compliance voltage – 20 V; TEC current ranges from -15 to +15 A; TEC compliance voltage > 15 V; TEC maxi-mum power > 225 W; controlled temperature ranges from -55 to +150 °C.”

Round 2

Reviewer 1 Report

The manuscript has been revised and improved.

Concerning the mechanical deficiencies of the Figures 2,3 and 5 we understand that the authors did some effort to increase the size of the axis and labels to improve the readibility without re-organizing the structure of the figures. It is worth noting that In the revised manuscript it seems that the old figures are still in the main text  

As a conclusion we guess that the paper could be now ready for publication.